# A Review of Receptor Recognition Mechanisms in Coronaviruses

**DOI:** 10.3390/v17121628

**Published:** 2025-12-16

**Authors:** Jie Liu, Wenjing Luo, Jianming Li, Bingyi Cai, Zhiwei Lei, Shiyun Lin, Zhuohong Chen, Zhaoyang Yue, Xulin Chen, Yongkui Li, Zhen Luo, Qiwei Zhang, Xin Chen

**Affiliations:** 1Institute of Medical Microbiology, Key Laboratory of Viral Pathogenesis & Infection Prevention and Control, Jinan University, Ministry of Education, Guangzhou 510632, China; liu938384@gmail.com (J.L.); luo19198027643@163.com (W.L.); li1026658422@gmail.com (J.L.); caiby2290910@outlook.com (B.C.); lsy15975026633@outlook.com (S.L.); zhluo18@jnu.edu.cn (Z.L.);; 2Affiliated Qingyuan Hospital, The Sixth Clinical Medical School, Guangzhou Medical University, Qingyuan People’s Hospital, Qingyuan 511500, China

**Keywords:** coronavirus, virus and host receptors interaction, spike protein, receptor mechanism

## Abstract

Cellular receptor recognition exerts fundamental roles during coronavirus infection. Clarifying the regulatory mechanism of virus receptor helps to better understand viral infection, transmission and pathogenesis; predict potential host and how viral escape from immune system; prevent coronavirus infection or develop treatment therapy. Herein, we summarize current understanding of host receptor recognition mechanisms in the different genera of coronavirus family. And we also review diverse methodologies of identification and clarification of virus receptors. The integration of structural biology, multi-omics, computational predictions, synthetic biology and artificially engineered viral receptors, provide a powerful framework for elucidating coronavirus–receptor interactions. This also supports the development of broad-spectrum antiviral agents, smart biosensors, and intervention strategies against emerging coronaviruses.

## 1. Introduction

Coronaviruses are positive-sense single-stranded RNA viruses that belong to the family Coronaviridae and the genus Coronavirus. They are named for the crown-like appearance under electron microscope. In humans, coronavirus infections range from mild, cold-like symptoms—including runny nose, fever, cough, and fatigue—to severe disease characterized by respiratory distress, viral pneumonitis, pneumonia, lung injury, and excessive inflammatory responses [1]. The infection can develop sever sickness, such as difficulty in breathing, acute viral pneumonitis, lung damage, pneumonia, hyperactive immune reaction and so on. Understanding the fundamental mechanisms of viral entry is essential for developing early antiviral interventions and reducing disease severity or prevent viral infection.

Coronavirus particles have an envelope structure and range in diameter from approximately 60 to 140 nanometers. Their positive-sense RNA genome spans 26–32 kb, the longest known among RNA viruses [2]. The genome of coronaviruses is highly variable and can evolve into new viral variants through mechanisms such as recombination, mutation, and rearrangement. According to the International Committee on Taxonomy of Viruses (ICTV), coronaviruses are classified into four genera: Alphacoronavirus, Betacoronavirus, Gammacoronavirus, and Deltacoronavirus [3]. Among them, Alphacoronaviruses and Betacoronaviruses primarily infect mammals, while Gammacoronaviruses and Deltacoronaviruses mainly infect birds.

The structure of coronaviruses mainly includes four key proteins:

Spike protein (S): Responsible for binding to host cell receptors and mediating viral entry into cells.

Membrane protein (M): Involved in viral assembly.

Envelope protein (E): Regulates viral budding.

Nucleocapsid protein (N): Binds to viral RNA and plays a role in replication and transcription.

The coronavirus spike protein (S protein) is an envelope glycoprotein located on the surface of the virus, forming the characteristic crown-like structure. It is a key molecule for the virus to recognize and fuse with host cells. This type I transmembrane glycoprotein consists of two subunits: S1, which is mainly responsible for receptor binding, and S2, which mediates membrane fusion. The S1 subunit contains two main domains: the N-terminal domain (NTD) and the C-terminal domain (CTD), both of which participate in binding to molecules on the host cell surface [4]. However, diverse types of coronaviruses use different receptor recognition mechanisms. Interaction between coronavirus S protein and the host receptor is a critical step for viral entry into cells. The receptor recognition process mainly involves three steps:

1. Virus-cell surface attachment: Virus entry into host cells is initiated by the attachment of viral particles to the cell surface. Virus attachment is frequently achieved through interactions with non-specific but highly expressed cell surface molecules, known as attachment factors. This process facilitates the accumulation and localization of the virus on the surface of target cells, thereby enhancing the efficiency of receptor recognition [5].

The glycocalyx is a layer of glycolipids and glycoproteins on the cell surface, and presents a range of molecules involved in viral attachment and entry. As glycocalyx mediates the interactions between viral surface and cell membrane, many viruses utilize carbohydrate molecules, such as heparan sulfate (HS), as attachment factors. Initially, viral particles anchor to the host plasma membrane via HS [6,7]. However, for successful infection, they must subsequently engage with protein receptors. As an attachment factor, HS enhances virus-cell interactions and promotes the opening of the receptor-binding domain (RBD) structure of the S protein, facilitating its binding to protein receptors. Additionally, sialoglycans (sialic acid-containing sugars) can also serve as initial attachment sites for certain coronaviruses. This process facilitates the accumulation and localization of the virus on the surface of target cells, thereby enhancing the efficiency of receptor recognition.

2. Receptor binding: Cellular receptors recognition and binding through viral particles is a prerequisite step for virus entry into host cells.

The spike (S) protein, located on the viral membrane, plays an important role in viral attachment and receptor binding (Figure 1 and Figure 2). Following initial attachment, the S1 subunit binds specific protein receptors through either its NTD or CTD. Receptor binding is a highly selective process and determines the host range and tissue tropism of the virus. Receptor engagement is influenced not only by molecular affinity, but also by dynamic conformational changes within the S protein, which regulates accessibility of its binding sites.

3. Membrane fusion: After receptor binding, coronaviruses enter the host cell through membrane fusion or endocytosis, releasing their genome into the host cell. Once the S1 subunit binds to the cellular receptor, it triggers conformational changes in the S protein, exposing the fusion peptide region within the S2 subunit, and preparing for subsequent membrane fusion [8]. The S2 subunit of S protein contains twoα-helical heptad repeat regions, HR1 and HR2, which assemble a stable six-helix bundle that promotes membrane fusion [9].

Cellular receptor recognition and virus entry are fundamental determinants of infectivity, transmission, and pathogenesis. Clarifying the regulatory mechanism of virus receptor improves our understanding of host susceptibility, viral immune evasion, and potential therapeutic targets for preventing or treating coronavirus infections.

## 2. Coronaviruses and Their Receptors

### 2.1. Alpha-Coronaviruses

Alpha-coronaviruses (Alpha-CoVs), are one of the four main genera in the coronavirus family, and primarily infect mammals, including humans, bats, cats, and dogs. Their broad host range and varied disease manifestations greatly affect interspecies transmission. High recombination rates and frequent point mutations drive genetic diversity, may enabling these viruses to cross species barriers and adapt rapidly to new hosts [10].

#### 2.1.1. Human Coronavirus HCoV-NL63

HCoV-NL63 was first isolated in 2004 from a 7-month-old child suffering from bronchiolitis and conjunctivitis. Its spike (S) protein is essential for recognizing and binding host cell receptors [11]. The N-terminal region of the S protein contains a unique 179-amino-acid domain [12]. The protein comprises two main subunits: S1 and S2. The S1 subunit includes a receptor-binding domain (RBD), which interacts with the angiotensin-converting enzyme 2 (ACE2) on the host cell surface, while the S2 subunit contains a hydrophobic fusion peptide and heptad repeat regions required for membrane fusions [13] (Figure 1 and Figure 2).

ACE2 is one of the major identified receptors of HCoV-NL63. The C-terminal domain (CTD) of the HCoV-NL63 S protein binds to ACE2. Variants with deletions in either the N-terminal 231 or C-terminal 57 amino acids revealed that residues 232 to 684 are involved in this interaction [14]. The NL63-CoV RBD binds the outer surface of the N-terminal helix of ACE2, engaging three distinct regions: VBM1 (residues 30–41), VBM2 (loop comprising residues 321–330), and VBM3 (loop comprising residues 353–356) [15]. During the binding of NL63, the Lys353 residue of ACE2 is embedded in a hydrophobic channel formed by Tyr498 and Ser535 of NL63, which facilitates the formation of a salt bridge between Lys353 and Asp38 on ACE2 [16]. The RBD of the HCoV-NL63 spike protein interacts with the VBMs of ACE2 through its three RBMs.

Heparan sulfate (HS) functions as an attachment receptor for HCoV-NL63, increasing viral density on the cell surface. Milewska et al. found that the S protein of HCoV-NL63 binds to cell surface heparan sulfate, facilitating viral adhesion and infection [17,18] (Figure 1 and Figure 2).

#### 2.1.2. Human Coronavirus HCoV-229E

HCoV-229E was first isolated in 1966 from students with respiratory illnesses. It shares high similarity with HCoV-NL63 within the alpha-coronavirus genus [19]. The S protein is critical for viral entry and membrane fusion (Figure 1 and Figure 2). The S protein includes 30 potential N-linked glycosylation sites. The S1 domain comprises residues 16–560, followed by an S2 domain with multiple heptad repeat regions, a transmembrane domain, and a cytoplasmic tail terminating with a cysteine residue.

Despite their similarities, HCoV-229E uses a different receptor, human aminopeptidase N (hAPN), for cellular entry (Figure 1 and Figure 2). A truncated S protein (S547) containing the N-terminal 547 residues binds hAPN expressed on 3T3 murine cells, with the interaction occurring between residues 417 and 547 in the C-terminal region of the S1 domain [20]. The S1 subunit consists of four beta domains arranged in a ring. Notably, interactions between residues 315–320 of loop 1 and residues 287–291 of hAPN are highly conserved. The side chain of Asn319 is buried within the binding interface, where it forms a hydrogen bond with Glu291 of hAPN. Additionally, a stacking interaction occurs between the backbone of Cys317 and the side chain of Tyr289 on hAPN. These interactions contribute to the stability of the binding between the HCoV-229E spike protein and hAPN. One of the domains, designated domain B (229E RBD), exhibits conformational flexibility; receptor binding occurs only when it adopts the “up” conformation [21].

The spike protein of HCoV-229E regulates its receptor-binding capacity through two distinct C1 and C2 conformational states. The transition from C1 to C2 reduces the inter-domain interaction surface within S1, resulting in RBD exposure. Concurrently, the fusion peptide, connecting region, and central helix within the S2 subunit tend to adopt a more open configuration in the C2 state, thereby priming the fusion peptide for exposure and facilitating S2′ cleavage [22].

N-glycosylation contributes to the spatial stability of the S protein and modulates receptor binding (Figure 1 and Figure 2). HCoV-229E and hAPN possess 30 and 10 predicted N-glycosylation sites, respectively. Among these, N62 on 229E displays the most prominent glycan density. High-density mannose-type glycans surrounding critical antigenic epitopes reduce neutralizing antibody recognition, thereby enabling immune evasion [23].

### 2.2. Animal Alpha-Coronaviruses

Phylogenetic analyses of zoonotic coronaviruses suggest that bats, rodents, or domesticated animals are the genetic sources of all seven known human coronaviruses (HCoVs). Notably, those HCoVs that cause the common cold are capable of spreading within the human population without requiring an intermediate animal host [24].

Therefore, investigating the receptors and potential cofactors of animal coronaviruses contributes to understanding receptor-binding capabilities, their influence on infection efficiency, and the possibility that different viral variants may exhibit altered tissue tropism based on changes in receptor affinity.

#### 2.2.1. Feline Coronavirus (FCoV)

Feline coronavirus (FCoV) is widespread among domestic cat populations globally. It typically causes mild gastrointestinal disease, but certain strains can lead to a fatal condition known as feline infectious peritonitis (FIP) [25]. FCoV is classified into two biotypes: feline enteric coronavirus (FECV) and feline infectious peritonitis virus (FIPV).

The spike (S) protein of FCoV, encoded by the S gene, is a type I transmembrane protein composed of three segments: an extracellular domain, a transmembrane domain, and a short cytoplasmic tail. The extracellular domain consists of the S1 receptor-binding domain (RBD) and the S2 membrane fusion domain [26]. A unique structural feature of the FCoV S protein is Domain 0, which is rotated 90° relative to the adjacent Domain A. This pronounced conformational difference is likely associated with glycosylation at residues N254, N357, and N491. The S2 subunit also contains multiple N-linked glycosylation sites. These glycan structures form a “glycan shield,” helping the virus evade recognition by the host immune system [27]. Although FECV and FIPV share high sequence similarity, specific mutations in the S protein, including Met-to-Leu at position 1058 and Ser-to-Ala at position 1060, have been identified in FIPV [28].

FCoV is further classified into serotypes I and II. Feline aminopeptidase N (fAPN), a membrane-bound metalloprotease, is widely expressed in intestinal epithelial cells, monocytes, fibroblasts, and other tissues [29]. Studies have shown that the S protein of type II FCoV interacts with specific regions of fAPN, R1 and R2, to facilitate viral entry. Among them, fAPN residues N740 and T742 play key roles in viral recognition. In contrast, type I FCoV does not use fAPN as its receptor. The precise receptor for type I FCoV remains unidentified, although glycan molecules such as sialic acid and heparan sulfate are thought to act as attachment factors to mediate initial viral binding [30] (Figure 1 and Figure 2).

Some research found that overexpression of human DC-SIGN in feline cells enhances FIPV infection. Inhibition of DC-SIGN pathway by anti-DC-SIGN antibodies or mannan, significantly reduce FIPV infectivity. These results let us keep in mind that human may be a potential host for FIPV [31].

#### 2.2.2. Transmissible Gastroenteritis Virus (TGEV) and Porcine Respiratory Coronavirus (PRCV)

Transmissible gastroenteritis virus (TGEV) is a coronavirus first discovered in 1946. It primarily causes severe diarrhea, dehydration, and high mortality in piglets, particularly those that are nursing [32].

The spike (S) protein of TGEV is cleaved by host proteases into two functional subunits: S1 and S2. The N-terminal S1 domain contains four antigenic sites and exhibits enterotropism, while the C-terminal region is essential for both enteric and respiratory tropism and for recognition of the aminopeptidase N (APN) receptor [33] (Figure 1 and Figure 2). The S2 subunit includes a fusion peptide, heptad repeats (HR1 and HR2), a transmembrane domain, and a cytoplasmic tail. These regions form the structural basis for fusion between viral and host cell membranes. During viral entry, HR1 and HR2 form a six-helix bundle (6-HB) that drives membrane fusion. PRCV is genetically similar to TGEV but has a large deletion in the N-terminal domain (NTD) of the spike protein, which impairs its ability to infect intestinal cells [34].

TGEV infection depends on recognition of host cell surface receptors [35]. The virus targets the VBM2 region of APN (residues 728–744). The β1–β2 region of the TGEV RBD interacts with the terminal region of the α19 and α21 helices of porcine APN (pAPN). The hydroxyl group of Tyr528 in the TGEV RBD forms hydrogen bonds with the side chains of Glu731 and Trp737 in pAPN, enhancing the binding affinity and specificity. Additionally, Gly527, adjacent to Tyr528, forms hydrogen bonds with the backbone of Asn736 in pAPN, contributing to the stability of the complex [36]. In the RBDs of PRCV and TGEV, the N-linked glycan at Asn736 of APN (NAG7361) interacts with Tyr528 of the viral RBD. The hydroxyl group of RBD Tyr528 forms hydrogen bonds with the side chains of Glu731 and Trp737 on pAPN, while the side chain of Gln530 forms hydrogen bonds with both NAG7361 and the side chain of Asn736 on pAPN.

TGEV can also bind to α-2,3-linked sialic acid residues on host cell surfaces via its spike protein NTD, thereby enhancing attachment to intestinal villous epithelial cells and facilitating initial infection (Figure 1 and Figure 2). This interaction allows the virus to target mucin-type glycoproteins such as MGP, helping it to overcome the mucosal barrier [37]. However, PRCV, as a variant of TGEV, lacks the ability to bind sialic acid and is thus unable to efficiently infect intestinal tissues [38].

#### 2.2.3. Porcine Epidemic Diarrhea Virus (PEDV)

PEDV primarily infects the porcine small intestine, where aminopeptidase N (APN) is abundantly expressed. pAPN can bind the S1 domain of the PEDV spike protein, and exogenous expression of pAPN has been shown to promote PEDV infection [39] (Figure 1). The spike (S) protein of PEDV shares structural similarities with that of TGEV, containing conventional domains such as the N-terminal domain (NTD or domain A) and the C-terminal domain (CTD or domain B). The APN-binding region of PEDV corresponds to the C-terminal half of the S1 domain (residues 477–629), which aligns with the receptor-binding domain (RBD) of TGEV S1 (residues 505–655) [40].Notably, PEDV’s S protein includes an additional Domain 0 (D0) located at the extreme N-terminus. Deletion of Domain 0 reduces the ability of PEDV to bind aminopeptidase N (APN), making the virus more dependent on the action of trypsin for cell entry [41]. However, research by Li W et al. revealed that APN is not essential for PEDV entry, as knockout of APN expression did not inhibit infection. This suggests that other cofactors may assist in receptor recognition [42].

Many studies have suggested that heparan sulfate (HS) could be a cofactor of PEDV (Figure 1). Enzymatic removal of HS using heparinase I inhibits PEDV infection, and disruption of HS biosynthesis by sodium chlorate also reduces infection efficiency [43]. These findings indicate that the precise receptor recognition mechanism of PEDV remains unclear and requires further investigation.

#### 2.2.4. Swine Acute Diarrhea Syndrome Coronavirus (SADS-CoV)

Swine acute diarrhea syndrome coronavirus (SADS-CoV) is an emerging alpha-coronavirus first identified during an outbreak in pig farms in 2017. SADS-CoV has been considered to originate from bats and possesses cross-species transmission potential [44]. The spike protein is divided into two subunits: S1, which is responsible for receptor binding, and S2, which mediates membrane fusion [45]. Structurally, the spike is composed of a β-sheet cap formed by the S1 subunit, a central stalk consisting of α-helices from the S2 subunit, and additional twisted β-sheets and loops from the S2 subunit [46].

So far, the specific receptor for SADS-CoV remains unidentified. However, Yang et al. reported that pretreatment of host cells with heparinase, which reduces the amount of cell-surface heparan sulfate (HS), significantly decreased SADS-CoV genomic infection levels. This suggests that the N-terminal domain (S1A) of the S1 subunit binds glycan receptors, such as sialic acid and HS, while the C-terminal domain (S1B) may be involved in recognizing protein receptors. Therefore, HS and sialic acid are considered important attachment cofactors for SADS-CoV [47] (Figure 1).

### 2.3. Betacoronaviruses

Betacoronaviruses (β-CoVs) are a group of coronaviruses that primarily infect mammals and represent the genus with the greatest impact on human health, including SARS-CoV, MERS-CoV, and SARS-CoV-2, which pose the most significant threats to humans Betacoronaviruses possess more complex genomic structures and stronger cross-species transmission capacity [48].

#### 2.3.1. Embecovirus (Lineage A)

##### Human Coronavirus OC43 (HCoV-OC43)

HCoV-OC43 is a common seasonal virus that generally causes mild upper respiratory tract infections. It was discovered in 1967 through human embryonic tracheal organ culture (OC) technology [49]. Studies have shown that HCoV-OC43 and bovine coronavirus (BCoV) exhibit significant antigenic and genetic similarities. Molecular clock analysis of the spike gene sequences of BCoV and HCoV-OC43 indicates that their most recent common ancestor dates back to around 1890, suggesting that BCoV crossed the species barrier to infect humans, possibly marking the origin of OC43 [50].

HCoV-OC43 belongs to lineage A, a small evolutionary branch of the Betacoronavirus genus. Unlike other coronaviruses, viruses in this lineage have two types of surface projections: one is a 20-nanometer spike—characteristic of coronaviruses—formed by trimeric spike proteins; the other is an 8-nanometer spike, unique to this evolutionary branch, composed of dimeric hemagglutinin-esterase (HE) proteins. HE is a sialate-O-acetylesterase with an additional 9-O-Ac-Sia-specific lectin domain, functioning as a receptor-destroying enzyme [51]. Before attachment, the receptor-destroying activity of HE prevents the virus from irreversibly binding to decoy receptors in the extracellular environment [52].

9-O-acetylated sialic acid (9-O-Ac-Sia) has been identified as one of the cellular entry receptors of HCoV-OC43 (Figure 1 and Figure 2). Binding experiments have shown that the S glycoprotein of HCoV-OC43 binds to O-acetylated sialic acid through domain A (S1A) on its S1 N-terminal domain (S1-NTD) [53]. The S1-NTD of the HCoV-OC43 spike protein contains a groove-like “glycan-binding pocket” structure that specifically recognizes 9-O-Ac-Sia. Structural studies indicate that this glycan-binding pocket interacts highly specifically with the acetyl group, forming multiple hydrogen bonds and hydrophobic interactions to ensure stable binding. However, the HE protein of HCoV-OC43, has lost the ability to bind 9-O-Ac-Sias because of its inactivated lectin domain [54].

A genome-wide CRISPR knockout screen identified aryl hydrocarbon receptor (AHR) as a key host dependency factor for HCoV-OC43 in the IGROV-1 cell line (Figure 2). DiMNF, a small-molecule AHR inhibitor, inhibited HCoV-OC43 infection in IGROV-1 cells [55]. Moreover, HLA class I molecules have been identified as one of the receptors of OC43. Many studies have reported that HCoV-OC43 could interact with HLA class I molecules on the cell surface to establish infection, which plays a role in the viral entry process [56] (Figure 1 and Figure 2).

##### Human Coronavirus HKU1 (HCoV-HKU1)

Human coronavirus HKU1 was first identified in 2005 in an elderly patient suffering from severe pneumonia [57]. HCoV-HKU1, like HCoV-OC43, belongs to lineage A of the Betacoronavirus genus, expressing two types of protrusions: spike (S) proteins and hemagglutinin-esterase (HE) proteins. The HE protein on the surface of HCoV-HKU1 has also lost the ability to bind 9-O-acetylated sialic acid (9-O-Ac-Sia), which appears to be an adaptation to the sialylated glycans of the human respiratory tract and replication within it.

O-acetylated sialic acid has been confirmed by several studies as an attachment factor for entry into host cells. It has been shown that the S protein of HCoV-HKU1 can specifically bind to the surface of the human rhabdomyosarcoma cell line RD. Pretreating RD cells with neuraminidase and trypsin significantly reduces this binding, suggesting that the interaction is mediated by sialic acids on glycoproteins [58]. Further structural analyses of the spike proteins of HCoV-OC43 and HCoV-HKU1 have shown that domain A on the S1 N-terminal domain (S1-NTD) is a key region responsible for binding 9-O-Ac-Sia (Figure 1). However, the receptor-binding site of the HCoV-HKU1 spike protein has significantly lower affinity for short sialylated glycans compared to that of HCoV-OC43, suggesting that the two viruses may differ in their fine specificity for receptor binding.

Some studies have indicated that, unlike other lineage A β-CoVs that use the N-terminal domain of S1 to engage with receptor proteins, HCoV-HKU1 may instead utilize the C-terminal domain (CTD) of its S1 subunit to bind to an as-yet-unidentified human receptor [59].

Transmembrane protease serine 2 (TMPRSS2) has been identified as one of a functional receptors for HCoV-HKU1 (Figure 1). TMPRSS2 exhibits high affinity for the receptor-binding domain of HCoV-HKU1. Anti-TMPRSS2 antibodies can effectively inhibit spike attachment, membrane fusion, pseudovirus based functional assay and viral infection of HCoV-HKU1 in primary human bronchial epithelial cells [60].

##### Mouse Hepatitis Virus (MHV)

Mouse Hepatitis Coronavirus (MHV) was first discovered in 1949 and is used as a model organism for studying coronaviruses. Its host range is limited to susceptible mice and mouse cell lines. MHV consists of a collection of strains with different organ tropisms, which can be divided into two main categories based on their tropism: one category is enterotropic, and the other includes polytropic strains, which target organs such as the nervous system, liver, and lungs [61].

The MHV S protein is cleaved by cellular proteases into two non-covalently bound subunits, the N-terminal S1 and C-terminal S2 [62]. X-ray crystallography studies revealed that the S1 subunit of MHV uses its N-terminal domain (NTD) as its receptor-binding domain. The core region of the MHV S1-NTD is a 13-strand β sandwich structure with two anti-parallel β sheets. This core structure is similar to the structure of human galactose-binding lectin, though despite having a galactose-binding lectin-like fold, MHV’s S1-NTD does not bind sugars. Instead, it forms a unique receptor-binding motif (RBM) in contact with the immunoglobulin-like domain of the N-terminal fragment of carcinoembryonic antigen-related cell adhesion molecule (CEACAM1) via three loops from the “upper” β-sheet of the NTD core, engaging in protein–protein interactions with mCEACAM1a [63]. Despite tropism differences, all MHV strains utilize the same receptor, carcinoembryonic antigen-related cell adhesion molecule 1 (CEACAM1) [64] (Figure 1 and Figure 2).

Furthermore, MHV’s S2 protein has been found not to participate in receptor binding with host cells [65]. Later studies identified mutations in the S protein of the MHV A59 strain that allowed the virus to enter cells in a heparan sulfate-dependent manner, making MHV less reliant on the CEACAM1a receptor and expanding its host range [66]. The variability of the S protein in MHV is not only a crucial strategy for the virus to adapt to its host and escape the immune system but also reflects one of the core mechanisms in the evolution of coronaviruses.

##### Bovine Coronavirus (BCoV)

Bovine coronavirus (BCoV) is a pathogen responsible for enteric and respiratory dis eases in cattle [67]. BCoV can spread rapidly via the fecal-oral and respiratory routes. Infection with BCoV leads to high mortality rates in calves, reduced growth rates in feeder cattle, and decreased milk production, leading to substantial economic losses for farmers [68].

Studies have shown that pretreatment of MDCK I cells with neuraminidase or acetylesterase renders the cells resistant to BCoV infection. Subsequent resialylation of the desialylated MDCK I cell with Neu5, 9Ac2 restores their susceptibility to BCoV infection. These findings indicate that BCoV recognizes 9-O-acetylated sialic acids on the cell surface as receptors for cellular entry [69,70].

As a lineage A coronavirus, BCoV also expresses a hemagglutinin-esterase (HE) protein on its surface. The HE protein consists of an esterase domain and a lectin domain, and it serves dual functions as both a lectin and a receptor-destroying enzyme (RDE), mediating reversible binding to O-acetylated sialic acids [71]. The lectin domain of HE facilitates viral particle binding, while also enhancing the activity of sialate-O-acetylesterase toward clustered sialylated glycans [72]. The RDE activity of the HE protein aids in the release of progeny virions and helps the virus evade attachment to non-permissive host cells [73]. Research further indicates that the spike (S) protein and the HE protein of coronaviruses are functionally interdependent and have co-evolved to optimize the balance between viral attachment and release [74].

#### 2.3.2. Sarbecovirus (Lineage B)

##### Severe Acute Respiratory Syndrome Coronavirus (SARS-CoV)

Severe acute respiratory syndrome (SARS) infection typically manifests with fever, cough, fatigue, and other influenza-like symptoms [75]. Genomic analysis showed that SARS-CoV was distinct from any of the three previously defined groups of coronaviruses [76].

The spike (S) protein of SARS-CoV contains two main functional subunits: the S1 subunit, which contains the receptor-binding domain (RBD) responsible for engaging the cellular receptor; and the S2 subunit, which primarily mediates membrane fusion between the virus and the host cell [77]. he C-terminal domain (CTD) of the S1 subunit is responsible for receptor binding. The S1-CTD of SARS-CoV contains two substructures: a core structure and a receptor-binding motif (RBM). The core consists of a five-stranded antiparallel β-sheet and several short α-helices, while the RBM features a slightly concave outer surface flanked by two ridges and a central antiparallel β-sheet used for binding to the cellular receptors, ACE2 for instance.

Several studies identified angiotensin-converting enzyme 2 (ACE2) as one of the cellular receptors for SARS-CoV [78] (Figure 1 and Figure 2). ACE2 is abundantly expressed in human lung and intestinal epithelium, providing potential routes for viral entry [79]. The extracellular domain of ACE2 comprises a membrane-distal peptidase domain and a membrane-proximal collectrin-like domain. The peptidase domain forms a claw-like structure with two lobes, and the active site of the enzyme lies within the cavity between these lobes [80]. The S1-CTD of SARS-CoV binds to the outer surface of the N-terminal lobe of ACE2, far from the enzyme’s active site, suggesting that SARS-CoV binding does not affect the enzymatic activity of ACE2. It has been shown that ACE2-mediated viral entry requires cell-surface heparan sulfate (HS) as a cofactor (Figure 1 and Figure 2). Removal of heparan sulfate proteoglycans (HSPGs) via heparinase treatment significantly inhibits SARS-CoV pseudovirus entry [81].

The receptor-binding domain of S protein is a critical determinant of tissue and host tropism. Mutations in the RBD of SARS-CoV played a key role in the virus’s cross-species transmission from civet cats to humans [82]. SARS-CoV also exhibits pH dependency mediated by S protein [83]. Furthermore, activation of the SARS-CoV S protein’s fusion capability requires cleavage by host cell proteases, such as the pH-sensitive cathepsin L [84].

##### Severe Acute Respiratory Syndrome Coronavirus 2 (SARS-CoV-2)

SARS-CoV-2 is a highly transmissible and pathogenic coronavirus that emerged in late 2019 and caused the COVID-19 pandemic. It is an enveloped, positive-sense single-stranded RNA virus sharing 79% nucleotide sequence identity with SARS-CoV [85]. Similarly to SARS-CoV, the spike (S) protein of SARS-CoV-2 consists of an S1 subunit containing the receptor-binding domain (RBD), and S2 subunit that mediates membrane fusion. Glycosylation of the SARS-CoV-2 spike protein plays a critical role in infection. Studies have shown that the glycan structures on the S protein surface can form a “glycan shield” that masks antigenic epitopes, reducing antibody recognition and thereby helping the virus evade the host immune system [86]. Variations in glycosylation patterns can also alter the strength of S protein binding to cell receptors [87].

A key distinction from other lineage B β-coronaviruses is the presence of a furin cleavage site at the S1/S2 boundary, which enhances the fusion capability of the Spike activation and likely contributes to the highly transmissibility of SARS-CoV [88].

The SARS-CoV2 spike protein mediates receptor engagement through the receptor-binding domain (RBD) located within S1. Following protease activation, RBD movement triggers a large-scale S1-to-S2 conformational rearrangement. S2 then refolds, exposing the fusion peptide and allowing the formation of heptad repeat 1 and 2 (HR1/HR2) six-helix bundles. The N-terminal region of HR2 adopts a single-turn helical configuration that closely packs into the groove of the HR1 coiled-coil, while the C-terminal region forms an extended helix that, together with the remainder of HR1, builds a second six-helix bundle. Progressive reinforcement along the longitudinal axis confers pronounced rigidity to the structure. These changes approximate viral and host membranes, open the fusion pore, and are complemented by strategic N-linked glycosylation after fusion [89,90].

ACE2 was rapidly identified as the primary cellular receptor for SARS-CoV-2 (Figure 1 and Figure 2). ACE2 is predominantly expressed in alveolar epithelial cells, the heart, kidneys, and intestines [91], which correlates with the organs affected during SARS-CoV-2 infection. The RBD of SARS-CoV-2 binds to ACE2 similar to that of SARS-CoV. Structurally, the RBD of SARS-CoV-2 features a twisted five-stranded antiparallel β-sheet core with several short α-helices, and its receptor-binding motif (RBM) has a slightly concave surface that accommodates the N-terminal helix of ACE2. Analysis of the SARS-CoV-2 RBD-ACE2 interface revealed that the SARS-CoV-2 RBD contacts 17 ACE2 residues, compared to 16 in SARS-CoV RBD. Of the 20 ACE2 residues involved in interacting with either RBD, 17 are shared, most of which reside on the N-terminal helix of ACE2. Among the 14 shared contact positions in the RBM, 8 have identical residues in both RBDs. Despite these similarities, differences in receptor recognition contribute to the significantly higher binding affinity of the SARS-CoV-2 RBD for ACE2 compared to that of SARS-CoV [92].

It has also been shown that the host serine protease TMPRSS2 is required for the activation of the SARS-CoV-2 S protein, facilitating viral entry into target cells [93]. After the host furin protease cleaves the S protein, a polybasic motif at the C-terminus of the S1 subunit is exposed. This motif can bind to neuropilin-1 (NRP1) on the cell surface, and blocking this interaction significantly reduces SARS-CoV-2 infectivity in cell culture, suggesting NRP1 may act as a co-receptor for the virus [94] (Figure 1 and Figure 2). Some studies have identified the tyrosine-protein kinase receptor UFO (AXL) as a potential co-receptor, due to its specific interaction with the N-terminal domain of the SARS-CoV-2 S protein, which may enhance viral infection [95] (Figure 1 and Figure 2). Additionally, similar to SARS-CoV, heparan sulfate proteoglycans (HSPGs) can function as viral attachment factors that assist the binding of the SARS-CoV-2 spike protein to ACE2 [96] (Figure 1 and Figure 2). A study has indicated that both SARS-CoV-2 and MERS-CoV can bind to DPP4, whereas SARS-CoV cannot. The former two share key binding residues of DPP4 at the interface. The insertion and substitution of E484 and its adjacent residues are critical factors contributing to the differential binding ability to DPP4 between the spike proteins of SARS-CoV-2 and SARS-CoV. DPP4 may serve as a candidate binding target for SARS-CoV-2 [97].

#### 2.3.3. Merbecovirus (Lineage C)

##### Middle East Respiratory Syndrome Coronavirus (MERS-CoV)

MERS-CoV causes a clinical syndrome similar to SARS, with symptoms such as fever, cough, and shortness of breath. Severe infection can lead to pneumonia, acute respiratory distress syndrome (ARDS), or multi-organ failure [98,99]. MERS-CoV strains isolated from camels are almost identical to those isolated from humans [100]. Retrospective studies found MERS-CoV antibodies in camel sera collected in 1983, indicating long-standing circulation in camels [101]. It is likely that MERS-CoV was transmitted to humans through an animal-to-human route.

The spike (S) protein of MERS-CoV is primarily divided into two functional regions: the S1 subunit, which contains the receptor-binding domain (RBD), with its core region located at amino acids 367–606 of the C-terminal region of the S1 subunit; and the S2 subunit, which mediates membrane fusion between the virus and host cell [102]. Similarly to the S1-CTD (C-terminal domain) of SARS-CoV, the S1-CTD of MERS-CoV also contains a core structure and a receptor-binding motif (RBM). Although their core structures are highly similar, the RBMs differ significantly, resulting in distinct receptor specificities between the two viruses [103]. The RBM of MERS-CoV’s S1-CTD consists of a four-stranded antiparallel β-sheet structure [104].

Dipeptidyl peptidase 4 (DPP4) has been identified as one of the cellular receptors for MERS-CoV (Figure 1 and Figure 2). DPP4 is highly conserved across different species [105]. DPP4 is a dimeric protein composed of two domains: an α/β-hydrolase domain and an eight-bladed β-propeller domain [106], the latter being the region where the viral RBD binds. The RBD of MERS-CoV S protein mainly interacts with the interface between blades 4 and 5 of DPP4’s β-propeller domain. The binding surface is relatively flat with subtle protrusions and depressions, and DPP4 engages the RBD through hydrophobic interactions and a network of hydrogen bonds. Notably, the receptor-binding domain of MERS-CoV interacts exclusively with the β-propeller domain of DPP4 and not with its enzymatic α/β-hydrolase domain, which explains why DPP4 enzyme inhibitors cannot prevent MERS-CoV entry into cells.

In addition, the S1A domain of MERS-CoV can bind to sialic acids on the host cell surface. Studies have shown that MERS-CoV preferentially binds to α2,3-linked sialic acids, enhancing viral attachment to host cells [107] (Figure 1). It has also been reported that the membrane-associated 78 kDa glucose-regulated protein (GRP78) is another binding target of the MERS-CoV spike protein (Figure 1 and Figure 2). Although GRP78 alone cannot render non-susceptible cells permissive to infection, it facilitates viral entry into susceptible cells by enhancing viral attachment [108]. Similarly, carcinoembryonic antigen-related cell adhesion molecule 5 (CEACAM5) has been identified as a cell-surface attachment factor for MERS-CoV. While CEACAM5 cannot mediate viral entry on its own, its overexpression enhances viral attachment and entry in susceptible cells, suggesting a synergistic role with DPP4 in promoting viral infection [109].

#### 2.3.4. Nobecovirus (Lineage D)

##### Bat Coronavirus HKU9 (BatCoV HKU9)

Bat coronavirus HKU9 is a representative lineage D betacoronavirus, currently known to infect only bats, with no evidence of human infection or pathogenicity.

No definitive functional receptor for HKU9 cellular entry has been identified. However, studies have indicated that GRP78 can interact with the HKU9 spike protein and promote viral adsorption on the cell surface. Nevertheless, GRP78 does not function as the primary receptor for viral entry; rather, it appears to act as a host factor that regulates or facilitates attachment [108]. Meanwhile, structural analysis of the HKU9 receptor-binding domain (RBD) has revealed an atomic structure composed of a core domain and an external subdomain. The core subdomain exhibits a folding pattern similar to that of other betacoronavirus RBDs, whereas the external subdomain features a distinctive single-helix structure. This structural uniqueness suggests that HKU9 may utilize a different or yet unidentified protein receptor for cell entry [110].

### 2.4. Gammacoronaviruses and Their Receptors

Gammacoronaviruses (Gamma-CoVs) primarily infect avian species and are among the most common coronaviruses found in poultry. The prototypical virus of this genus is the infectious bronchitis virus (IBV) [111].

#### Infectious Bronchitis Virus (IBV)

IBV is a coronavirus that exclusively infects avian species, particularly chickens, and mainly causes acute upper respiratory tract infections. The spike (S) protein of IBV consists of two subunits: S1, which is responsible for receptor recognition, and S2, which mediates membrane fusion [112]. The S1 subunit contains two major structural domains—S1-NTD and S1-CTD—as well as two subdomains, SD1 and SD2. The core structure of the S1-NTD is a 12-stranded β-sandwich formed by two anti-parallel β-sheets, which are stacked together through hydrophobic interactions [113]. Cryo-EM analysis has revealed the folding features of the infectious bronchitis virus (IBV) spike protein and the conformational characteristics of its receptor-binding head (S1). Compared with α- and β-coronaviruses, the S1 heads of γ-coronaviruses display markedly higher sequence and structural diversity, which accommodates distinct glycan-binding properties and may facilitate the recognition of potential glycan-binding sites and antigenic surfaces [114].

To date, the natural host cell receptor for IBV remains unclear. However, Winter et al. found that infection of primary chicken kidney cells by the IBV M41 strain depends on the presence of sialic acid. Amino acids 19 to 69 within the S1-NTD preferentially recognize α2,3-linked sialic acid residues, suggesting that sialic acid acts as an important cofactor in IBV infection [115] (Figure 1).

### 2.5. Deltacoronaviruses and Their Receptors

Deltacoronaviruses (Delta-CoVs) represent a newly identified group within the coronavirus family, that primarily infects avian species and mammals such as pigs and cattle. Research has mainly focused on the porcine deltacoronavirus (PDCoV).

#### Porcine Deltacoronavirus (PDCoV)

PDCoV was first identified in 2012 during surveillance of coronavirus in mammals [116]. Like other coronaviruses, the spike (S) protein of PDCoV is composed of S1 and S2 subunits. The S1-NTD of PDCoV adopts a β-sandwich fold similar to that found in alpha- and beta-coronavirus S1-NTDs [117].

PDCoV uses aminopeptidase N (APN) as its entry receptor. The S protein interacts with APN via its domain B, also referred to as the receptor-binding domain (RBD) [118]. The S1 subunit of PDCoV contains two receptor-binding domains: S1A and S1B. Liu et al. demonstrated that hemagglutinin (HA)-tagged porcine APN (pAPN) binds the PDCoV S1 protein, and immunostaining confirmed that pAPN on the cell surface interacts with the S1B domain of PDCoV S1 [119]. The porcine deltacoronavirus (PdCoV) spike protein possesses several structural features that may contribute to immune evasion, including a reduced solvent-exposed surface area, masking of receptor-binding sites, and shielding of key S1 epitopes. The compact organization of the PdCoV spike lowers its overall surface accessibility. After host–cell engagement, the PdCoV S1-CTD undergoes a transition from a closed to an open conformation, enabling the putative receptor-binding motif (RBM) loop to interact with host receptors [117]. PDCoV can infect a wide range of avian and mammalian cells. It recognizes conserved APN residues among human and porcine (Figure 1 and Figure 2). Its receptor-binding domain (RBD) interacts with domain II (DII) and domain IV (DIV) of APN. Within domain II, residues D317, F318, and E320 of the PDCoV RBD form hydrogen bonds and salt bridges with residue K379 of APN. In domain IV, residues N397, Y398, L399, L400, and R401 of the RBD establish extensive hydrogen bonding and salt bridge interactions with the α19–α20 loop of APN, comprising residues R741–E742–I743–P744–E745 [120].

Heparan sulfate (HS) also assists PDCoV binding and infection (Figure 1 and Figure 2). Studies have shown that HS enhances cell attachment of PDCoV, while reduction in HS sulfation using sodium chlorate inhibits viral binding and infection [121]. Therefore, HS serves as an attachment factor for PDCoV.

## 3. Discussion

Recent breakthroughs in structural biology techniques have allowed researchers to precisely resolve the dynamic conformational changes in the receptor-binding domains (RBDs) of the S proteins and the molecular details of their binding to the receptors. We are now discussing the methods and significance of coronavirus receptor identification, including some unknown or emerging coronaviruses. These findings have pushed coronavirus research from reactive defense to proactive intervention.

### 3.1. Methodology of Coronavirus Receptor Identification

Based on the principles of virus–host interactions, multiple techniques, such as reverse genetics systems, gene editing, antibody blocking, and structural analysis of viral and host proteins, have been applied as a strong theoretical foundation for identifying known and novel coronavirus receptors (Figure 3).

Multiple techniques are used to cross-validate receptor function in viral entry step. Interaction assays, such as affinity purification-mass spectrometry (AP-MS) Virus Overlay Protein Binding Assay (VOPBA), and surface plasmon resonance (SPR), are applied to screen and analyze host–virus interactions Structural studies with cryo-electron microscopy (Cryo-EM) and other techniques, contribute to understanding the structure of virus and host receptors. Functional assays, such as CRISPR-Cas9 genome-wide screening [122], viral gene deletion systems (e.g., S or RBD deletions), or humanized animal models (e.g., ACE2 humanized mice) [123] have been applied to confirm the critical role of receptors in viral entry. Recent breakthroughs in structural biology techniques such as cryo-EM single-particle analysis, cryo-ET and X-ray crystallography help researchers to better characterize the structure of virus S protein-receptor complexes, dynamic conformational changes, and the binding mechanisms of receptor complexes (Table 1).

A systematic strategy for identifying unknown coronavirus receptors begins with evolutionary analysis of the spike protein (S protein). Through multi-sequence alignment and phylogenetic studies, conserved regions and key variant sites within the receptor-binding domain (RBD) could be identified [131]. In addition, phylogenetic analysis of the S protein can distinguish potential receptor preferences among different coronavirus genera (e.g., α and β genera) [132]. These analyses provide targeted directions for subsequent receptor screening. From bioinformatics prediction to in vivo validation, it can provide a feasible methodology for reveal unknown coronavirus receptors.

### 3.2. The Significance of Viral Receptors Research

#### 3.2.1. Elucidating the Regulatory Mechanisms of Viral Receptors Is Help to Predict Potential Host Cells and Host Species

Better understand interspecies and cross species transmission helps to prevent virus spread. studies on coronavirus receptor-binding interfaces provide critical evidence for predicting potential host ranges and cross-species transmission risks. By integrating cryo-EM structural analysis with AI prediction tools (e.g., AlphaFold2), researchers can rapidly analyze interaction patterns between viral S proteins and receptor proteins across species, evaluating their binding compatible. For instance, the binding of HKU5-CoV-2 with human ACE2, at lower affinity suggests limited interspecies transmission risk. The structural-functional analysis, combined with phylogenetic tree, enables systematic evaluation of viral receptor utilization patterns in potential hosts; and provide biological markers for viral spread warning [133]. Future efforts should establish cross-species receptor databases and integrate single-cell sequencing technologies to map receptor expression profiles across animal tissues. It could help to accurately predict viral host-jumping ability. Importantly integration of multi-discipline studies could boost the identification of host receptors for emerging viruses. Combination of cryo-electron microscopy with AI tools also could accelerate the structural elucidation of receptor-virus complexes, which can further reveal the conservation and variation patterns of key binding sites. Simultaneously, machine learning models predicting receptor binding affinity, coupled with virus infection assays, enable rapid screening of high-risk viral strains.

#### 3.2.2. Elucidating the Regulatory Mechanisms of Viral Receptors Is Critical for Understanding Viral Transmission and Escape

Coronaviruses (CoVs) have drawn significant attention due to their extensive potential reservoirs and complicated transmission mechanisms. So far, the cross-species transmission of coronavirus remains unknown. The interaction of coronavirus with host receptor may play key roles in cross-species transmission. Analysis of APN orthologs from 17 species, evaluation their binding affinity with TGEV, and TGEV infection ability in BHK-21 cells, together with structure studies of RBD-dAPN interaction, sheds light on the potential cross-species transmission of two porcine CoVs [134]. In addition, analysis of ACE2 orthologs from different host species and stably expressing ACE2 orthologs in SARS-CoV-2 infected-A549 cells highlights a potentially broad host tropism of SARS-CoV-2 [135].

Investigating these potential reservoirs is vital for disease prevention, control, and therapeutic development [136]. Studying the receptor mechanism of coronavirus is to better understand how viral escapes from immune system. Taking the SARS-CoV-2 Omicron variant as an example, the S protein ingeniously evades immune system recognition through frequent recombination and mutation. Antigenic drift occurs frequently, such asN501Y and E484K mutations the receptor-binding domain (RBD) of the S protein, which substantially reduces antibody neutralization efficiency [137]. Furthermore, the furin cleavage site in the spike protein, exemplified by the SARS-CoV-2 PRRA insertion, not only enhances viral infectivity, but also successfully evades innate immune defenses. To uncover these complex mechanisms, researchers develop several techniques, such as single-cell sequencing and animal models (e.g., ACE2 transgenic mice). to reveal the molecular basis for viral latency and reactivation.

#### 3.2.3. Elucidating the Regulatory Mechanisms of Viral Receptors Contribute to Preventing Coronavirus Infection or Developing Therapeutic Approaches

Based on the virus–host receptor interaction mechanisms, vaccines can be designed to deliver the key viral surface molecules or highly conserved antigens to induce broad immune protection. As coronaviruses continue to mutate, bivalent and multivalent vaccines have emerged. Their antigen combinations targeting multiple variants significantly broaden the scope of immune protection.

Current therapeutic strategies for coronaviruses primarily target two critical pathways: blocking viral entry and inhibiting replication. In antibody therapies, monoclonal antibodies (e.g., Casirivimab/Imdevimab) block viral binding to ACE2 by targeting the RBD, though their efficacy is susceptible to variant mutations (e.g., Omicron’s K417N/E484K mutations) [138]. Broad-spectrum neutralizing antibodies targeting conserved regions of the S protein (e.g., the S2 stem) and soluble ACE2 receptor decoy (APN01) demonstrate broader antiviral potential. Small-molecule inhibitors targeting viral-receptor surface are also developed quickly and are being conducted in clinical trials [139], such as TMPRSS2 inhibitors (e.g., Camostat) that block viral membrane fusion [93]. Moreover, multivalent nanomedicines with dual virus-blocking and immune-modulating functions could be an efficient way to counter viral escape mutations [140]. Neutralizing monoclonal antibodies include sotrovimab (GSK4182136 or S309), bamlanivimab-etesevimab [141] (LY-CoV016-LY-CoV555), and casirivimab-imdevimab (REGN-CoV2) [142], which prevent infection of human cells by blocking the S-protein–ACE2 attachment and mediate SARS-CoV-2 entry into the human respiratory epithelial cells. These antibody therapies have been authorized by the FDA for COVID-19 treatment options. A neutralized bispecific antibody BM219 targets two conserved epitopes on the SARS-CoV-2 spike protein, which effectively neutralizes multiple variants, including JN.1. Clinical studies further demonstrate that it rapidly reduces respiratory viral load in infected individuals and shortens the median duration of symptoms [143]. Certain spike protein inhibitors, such as Griffithsin, have also been tested for clinical application [144].

Translational research based on receptor recognition mechanisms has had a profound impact on the prevention and control of coronaviruses. This impact has been felt across three key dimensions: therapeutics, vaccines, and proactive early warning systems. Therapeutic strategies are centered on the overcoming of viral escape by targeting conserved epitopes and the development of novel antibodies and inhibitors. Vaccine design employs multivalent and chimeric antigen strategies to induce broader cross-immunity protection. Proactive surveillance employs a multifaceted approach that integrates computational modeling and artificial intelligence to predict viral evolution and guide active monitoring. This integrated framework denotes a transition in the management of the virus from reactive measures to precision design and proactive intervention, grounded in molecular mechanisms.

Advances in computational prediction and structural biology have accelerated the design of engineered receptors and antiviral candidates. Yan Huan’s team designed functional receptors for multiple coronaviruses by reconstructing and optimizing customized virus receptors (CVRs). The team’s engineered receptor S2L20-CVR activates viral fusion mechanisms through an ACE2-independent, non-canonical binding site. This provides modeling support for studying viruses with unknown receptors and accelerates the development of antiviral drugs and vaccines [145]. Intelligent drug design platforms can predict RBD dynamic conformations via deep learning and develop novel nanobodies with molecular anchoring functions. Integration of AI tools, such as AlphaFold2, cryo-EM modeling and Intelligent drug design platforms, can design personalized block receptors or peptides to inhibit viral infection.

#### 3.2.4. Emerging Technologies for Studying Coronavirus Receptor Recognition

With the integration of emerging technologies, large-scale single-cell RNA-sequencing (scRNA-seq) datasets enable the identification of co-expression patterns of receptors and entry cofactors across tissues and under distinct developmental or pathological conditions, providing a higher-resolution framework for studying coronavirus receptor biology. For example, scRNA-seq has revealed the cell-type-specific expression of key receptors such as ACE2, and comparative single-cell atlases across tissues and populations allow the identification of highly susceptible cell types [146].

In the realm of artificial intelligence, AI models incorporating deep mutational scanning (DMS) data and structure-based prediction can forecast the binding effects of newly emerging mutations, supporting the assessment of viral transmissibility and immune-escape potential. AI-assisted analysis can prospectively identify high-risk RBD mutations and potential alternative receptors, thereby accelerating neutralizing-antibody design and strengthening public-health surveillance [147]. For understanding receptor recognition across the numerous future coronavirus variants, an important direction is to integrate high-resolution cellular atlases with AI-based structural and functional prediction, linking molecular interactions to cellular and tissue-level susceptibility.

## 4. Outlook and Challenges

To prepare for the future, it is imperative to establish a “prototype pathogen” research framework—constructing a database of common targets based on existing coronavirus libraries and developing modular drug design platforms [148]. Concurrently, global data sharing and interdisciplinary collaboration (e.g., structural biology, computational chemistry, and immunology) will become core safeguards against unknown viruses [149]. Establish virus mutation database using deep learning tools and integrating multi-omics data (proteomics, metabolomics), help to identify high-risk variants and monitor animal coronavirus reservoirs. Future studies are more incline to focus on precisely mapping virus and host interactions at single-cell level and single-atom resolution. Clarifying the virus–host receptor mechanism can also help to exploit vaccine, drug or antiviral micro-robots. These studies will develop a flourishing future for more techniques and equipment (Figure 4).

## Figures and Tables

**Figure 1 viruses-17-01628-f001:**
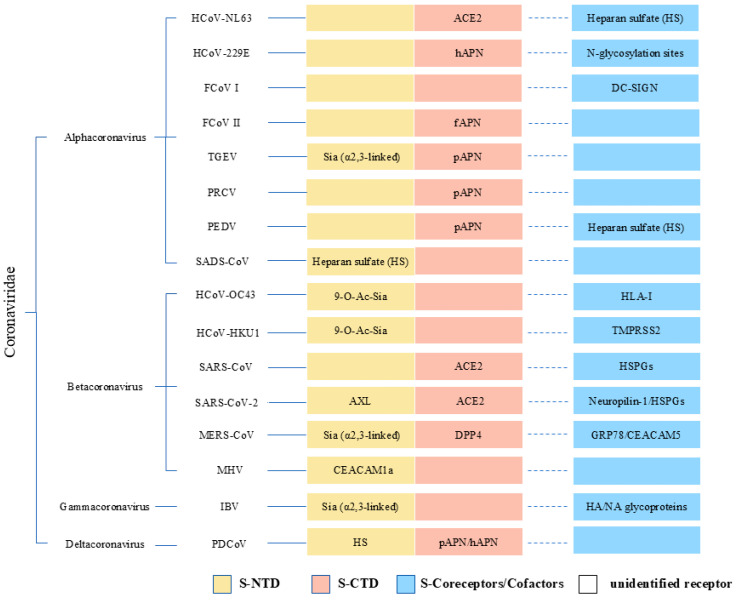
**Receptor binding of different structural domains of the coronavirus spiking protein (S).** This figure systematically shows the genetic relationship of the four genera of coronaviruses, α, β, γ, and δ. The solid line of differentiation labels the primary receptor (NTD: yellow,CTD:pink), and the dashed line of differentiation indicates co-receptors or cofactors (blue). And blank space represents unidentified receptors.

**Figure 2 viruses-17-01628-f002:**
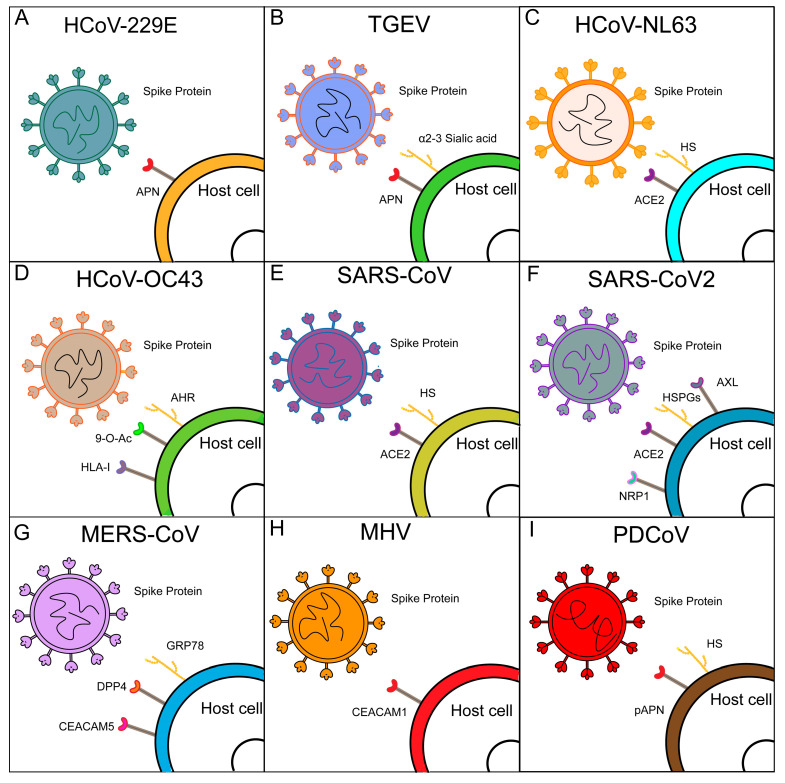
**Schematic representation of the recognition and binding of host cell receptors by spike (S) proteins of different coronaviruses.** (**A**) HCoV-229E utilizes APN as a receptor for cell-surface recognition. (**B**) TGEV recognizes APN receptor, while α-2,3-linked sialic acid functions as a cofactor to facilitate infection. (**C**) HCoV-NL63 binds to ACE2 receptor, and HS acts as cofactors. (**D**) HCoV-OC43 recognizes the cell-surface receptor 9-O-Ac, and interacts with the cofactor HLA-I and AHR to facilitate infection. (**E**) SARS-CoV employs ACE2 as a receptor, and HS functions as a cofactor to mediate cell-surface recognition. (**F**) SARS-CoV-2 interacts with ACE2 receptor, and the auxiliary receptors NRP1, HS, and AXL promote infection. (**G**) MERS-CoV recognizes DPP4 as a receptor, and the cofactors CEACAM5 and GRP78 to enhance cell infection. (**H**) MHV uses CEACAM1 as a receptor. (**I**) PDCoV recognizes pAPN as a receptor for cell infection, with HS acting as a cofactor for cell-surface recognition.

**Figure 3 viruses-17-01628-f003:**
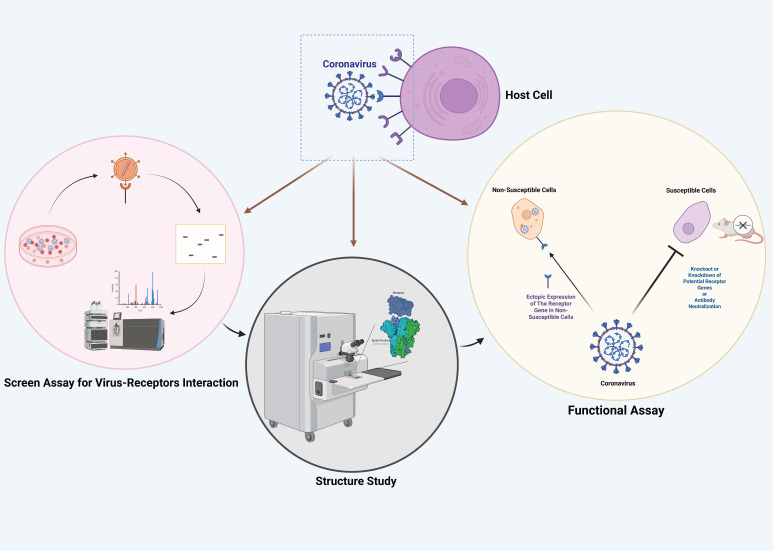
**Strategies for identifying coronavirus-interacting host receptors.** (**Left panel**) Identification of virus and receptor interactions by several techniques, such as immunoprecipitation and mass spectrometry, are used to screen and identify candidate receptor molecules, quantify interactions between viral surface proteins and potential host receptors. (**Middle panel**) Structure studies focus on elucidating the molecular architecture of the coronavirus–receptor complex using advanced structural biology techniques. (**Right panel**) Functional assay for receptor validation. Knockout/knockdown of receptor genes or receptor antibody neutralization experiments in susceptible cells or animal models to analyze viral infection capability.

**Figure 4 viruses-17-01628-f004:**
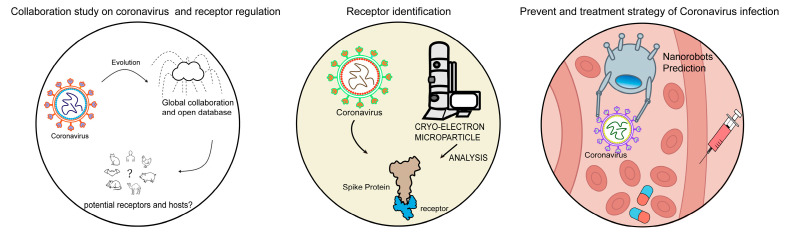
**The significance and outlook of receptor study in Coronaviruses.** (**Left panel**) global collaboration and database sharing are important for studying viral evolution, potential receptors, and host species. (**Middle panel**) Analyzing viral spike protein and receptor complex structure is crucial for understanding receptor-binding mechanisms and host specificity. (**Right panel**) Understanding receptor regulatory mechanisms helps to develop prevention or treatment strategies of coronavirus infection in the future, such as vaccine development, drug discovery, and nanorobotic prediction systems in antiviral therapy.

**Table 1 viruses-17-01628-t001:** Technologies applied in Receptor Identification.

Technologies	Advantages	Typical Applications Limitations	Typical Applications
Cryo-Electron Microscopy (Cryo-EM)	Samples remain in their “native state”; high-resolution structures can be obtained; dynamic structures of biomolecules can be studied	High demands on sample purity and homogeneity; expensive equipment	Three-dimensional structure of the SARS-CoV-2 S protein complexed with ACE2 receptor [124].
CRISPR-Cas9 Screening	Enables unbiased genome-wide screening to discover novel host factors; exhibits high specificity and efficiency	Risk of off-target effects	Identification host factors for SARS-CoV-2, gama- and delta-coronavirus infection [125,126]
AlphaFold2	Rapidly predicts protein 3D structures from amino acid sequences alone	Limited prediction of protein conformational dynamics; predicted structures may not perfectly match experimental structures	Predicts SARS-CoV-2 E protein structures and potential conformational states [127]. predict interactions between multiple β-coronavirus nucleocapsid proteins and human cytokines [128].
Affinity Purification Mass Spectrometry (AP-MS)	Systematically captures entire protein complexes interacting directly or indirectly with target proteins under near-physiological conditions.	May miss weak/transient interactions; requires subsequent validation.	Systematically screens host cell membrane protein complexes that interact with SARS-CoV-2 spike protein. [129]
Surface Plasmon Resonance (SPR)	Enables real-time, label-free, precise measurement of binding kinetics and affinity for molecular interactions	Requires purified proteins; cannot fully simulate the cell membrane environment.	Quantitatively determines the affinity and kinetic parameters of viral protein binding to host receptors [130].

## Data Availability

No new data were created or analyzed in this study.

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
