# Peer review of "A Review of Receptor Recognition Mechanisms in Coronaviruses"

_viruses, 2025, doi:10.3390/v17121628_

Round 1
Reviewer 1 Report
Comments and Suggestions for Authors
This article reviews the research progress on the receptor recognition mechanism of coronaviruses, emphasizing the powerful framework role of various methods such as structural biology, multi-omics, computational prediction, synthetic biology, and artificial engineering virus receptors in clarifying the interaction between coronaviruses and their receptors. These studies not only contribute to understanding the infection, transmission, and pathogenic mechanisms of coronaviruses but also provide support for the development of broad-spectrum antiviral drugs, intelligent biosensors, and intervention strategies against emerging coronaviruses.
- Some contents can be discussed in depth, such as the diversity of the virus receptors. Why are the receptors of SARS-CoV and SARS-CoV-2 both ACE2, while the receptor of MERS is DPP4, even though they are all beta-coronaviruses?
- Receptors play a crucial role in cross-species transmission, and the structural diversity of receptors may influence the host range and tissue tropism of viruses. In-depth discussion of these diversities can help understand the evolution and transmission mechanisms of viruses. Can examples be provided to illustrate which coronaviruses have undergone cross-species transmission due to similar receptors between species? Which species are at risk of spillover infection due to similar receptor structures? There is relatively little analysis of the evolution of viral receptors in the literature. It is suggested that the authors add a detailed discussion on the changes in coronavirus receptors during the evolutionary process, especially by comparing the receptor sequences of different coronavirus genera to reveal their evolutionary paths and cross-species risks.
- 2.3. Betacoronaviruses Among β-coronaviruses, many animal coronaviruses have also identified receptors, including those from bats, pangolins and bovine coronaviruses, etc. The authors should conduct a further review.
- There is relatively little discussion in the literature regarding the clinical application of viral receptors. It is suggested that the authors add a detailed discussion on the application of viral receptors in clinical diagnosis and treatment, especially for the diagnosis and treatment strategies of SARS-CoV-2. By discussing these applications, practical information can be provided for the development of target drugs.
- There is also little discussion about the future research directions of coronavirus receptors. The authors could add a detailed discussion on the future research directions of the recognition mechanism of coronavirus receptors, especially the application prospects of emerging technologies (such as single-cell sequencing and artificial intelligence) in this field.
- It chould be provided more detailed explanations and illustrations for the key mechanisms of coronavirus receptor recognition, such as the conformational changes of the S protein and the membrane fusion process. Detailed explanations can help readers better understand the complex biochemical processes, especially for those who are not familiar with this field.
- When discussing the receptor recognition mechanisms of different coronaviruses, comparative analyses could be added. Comparative analyses can help readers better understand the characteristics and evolutionary relationships of different coronaviruses, and demonstrate the similarities and differences among different viruses.
- Figures 1-4 are not cited in the text.
Author Response
Comments 1. Some contents can be discussed in depth, such as the diversity of the virus receptors. Why are the receptors of SARS-CoV and SARS-CoV-2 both ACE2, while the receptor of MERS is DPP4, even though they are all beta-coronaviruses?
Response 1:Thank you for pointing this out. Therefore, we have searched for all the studies of SARS-CoV, SARS-CoV-2 and MERS-CoV’s host receptors. As SARS-CoV and SARS-CoV-2 are classified as sarbecoviruses, a subgenus within the betacoronavirus family; while MERS-CoV is classified as merbecovirus, it is very likely that different receptor usage between these two-subgenus coronaviruses [1]. Li et al. demonstrated that the MERS-CoV Receptor DPP4 as a Candidate Binding Target of the SARS-CoV-2 Spike [2]. The E484 residue in the RBD region of the S protein is likely an important determinant affecting the binding of DPP4. As DPP4 was not able to mediate the SARS-CoV-2 entry independently in nonpermissive cells shown by Hoffmann et al., 2020 [3], Letko et al., 2020 [1], Zhou et al., 2020 [4]. These data suggest that ACE2 may be the primary receptor for SARS-Cov and SARS-Cov-2 infections, while DPP4 may serve as a potential co-receptor for both viruses. To date, no cases of MERS-COV infection have been reported in individuals possessing the ACE2 receptor. However, MERS-COV-like viruses are capable of binding to the ACE2 receptor, which serves as their entry mechanism into the host cell.
We have clarified these in the ‘2.3 Betacoronaviruses’ of our revised manuscript (See Page 10, line 468-473).
Comments 2. Receptors play a crucial role in cross-species transmission, and the structural diversity of receptors may influence the host range and tissue tropism of viruses. In-depth discussion of these diversities can help understand the evolution and transmission mechanisms of viruses. Can examples be provided to illustrate which coronaviruses have undergone cross-species transmission due to similar receptors between species? Which species are at risk of spillover infection due to similar receptor structures? There is relatively little analysis of the evolution of viral receptors in the literature. It is suggested that the authors add a detailed discussion on the changes in coronavirus receptors during the evolutionary process, especially by comparing the receptor sequences of different coronavirus genera to reveal their evolutionary paths and cross-species risks.
Response 2:Thank you for pointing this out. Therefore, we have searched all the literatures reporting possible cross-species transmission of coronavirus. So far, the cross-species transmission of coronavirus remains unknown. The interaction of coronavirus with host receptor may play key roles in cross-species transmission. Analysis of APN orthologs from 17 species, evaluation their binding affinity with TGEV, and TGEV infection ability in BHK-21 cells, together with structure studies of RBD-dAPN interaction, shed light on the potential cross-species transmission of two porcine CoVs [5]. In addition, analysis of ACE2 orthologs from different host species and stably expressing ACE2 orthologs in SARS-CoV-2 infected-A549 cells highlight a potentially broad host tropism of SARS-CoV-2 [6].
We have clarified these in the ‘3.2.2. Elucidating the Regulatory Mechanisms of Viral Receptors Is Critical for Understanding Viral Transmission and Escape’ of our revised manuscript (See Page 14, line 642-649).
Comments 3. 2.3. Betacoronaviruses Among β-coronaviruses, many animal coronaviruses have also identified receptors, including those from bats, pangolins and bovine coronaviruses, etc. The authors should conduct a further review.
Response 3: Thank you for pointing this out. According to the comment, we revised the coronavirus section into four subgenera based on its classification, and we added an introduction to the receptor recognition mechanisms of bat coronaviruse HKU9 and bovine coronavirus.
Please see the ‘2.3.4 Nobecovirus (lineage D)’ section of our revised manuscript (See Page 11, line 520-529).
Comments 4. There is relatively little discussion in the literature regarding the clinical application of viral receptors. It is suggested that the authors add a detailed discussion on the application of viral receptors in clinical diagnosis and treatment, especially for the diagnosis and treatment strategies of SARS-CoV-2. By discussing these applications, practical information can be provided for the development of target drugs.
Response 4: Thank you for pointing this out. We have added a detailed discussion on the application of viral receptors in clinical diagnosis and treatment in the ‘3.2.3 Elucidating the Regulatory Mechanisms of Viral Receptors Contribute to Prevent Coronavirus Infection or Developing Therapeutic Approaches’ (See Page 15, line 679-689).
Comments 5. There is also little discussion about the future research directions of coronavirus receptors. The authors could add a detailed discussion on the future research directions of the recognition mechanism of coronavirus receptors, especially the application prospects of emerging technologies (such as single-cell sequencing and artificial intelligence) in this field.
Response 5: Thank you for the good suggestion. We have supplemented our work with discussions on emerging technologies such as single-cell sequencing and artificial intelligence, as well as the future research directions for elucidating coronavirus receptor recognition mechanisms. Please see the ‘3.2.4 Emerging Technologies for Studying Coronavirus Receptor Recognition’ of our revised manuscript (See Page 15, line 713-729).
Comments 6. It chould be provided more detailed explanations and illustrations for the key mechanisms of coronavirus receptor recognition, such as the conformational changes of the S protein and the membrane fusion process. Detailed explanations can help readers better understand the complex biochemical processes, especially for those who are not familiar with this field.
Response 6: Thank you for the good comment. The primary mechanism underlying coronavirus receptor recognition is mediated by conformational changes in the viral spike (S) protein, which subsequently enable receptor engagement. Distinct coronavirus genera, including α-, β-, δ-, and γ-coronaviruses, exhibit divergent S-protein conformational dynamics. Each of which plays a critical role in receptor binding and membrane fusion. We now characterize the genus-specific conformational transitions of the S protein and their impact on the fusion process in the ‘2. Coronaviruses and Their Receptors ’ of our revised manuscript (See Page 4, line 150-155,Page 10, line 433-443, Page 12, line 542-547 Page 12, line 567-573).
.
Comments 7. When discussing the receptor recognition mechanisms of different coronaviruses, comparative analyses could be added. Comparative analyses can help readers better understand the characteristics and evolutionary relationships of different coronaviruses, and demonstrate the similarities and differences among different viruses.
Response 7:Thank you for the good comment. We illustrate the similarities and differences in the recognition of different coronavirus receptors, analyzing the similarities and differences in the types of receptors recognized by distinct coronaviruses, we provide a more intuitive depiction of the features within their receptor-binding regions, as well as the variations among the receptors utilized by different coronavirus species. Please see the Figure 1 and 2 of our revised manuscript.
Comments 8. Figures 1-4 are not cited in the text.
Response 8: Thank you for point out this problem. We have now cited the Figure 1-4 in the text.
Reference
(1) Letko, M.; Marzi, A.; Munster, V. Functional assessment of cell entry and receptor usage for SARS-CoV-2 and other lineage B betacoronaviruses. Nat Microbiol 2020, 5, 562-569, https://doi.org/10.1038/s41564-020-0688-y.
(2) Li, Y.; Zhang, Z.; Yang, L.; Lian, X.; Xie, Y.; Li, S.; Xin, S.; Cao, P.; Lu, J. The MERS-CoV Receptor DPP4 as a Candidate Binding Target of the SARS-CoV-2 Spike. iScience 2020, 23, 101160, https://doi.org/10.1016/j.isci.2020.101160.
(3) Hoffmann, M.; Kleine-Weber, H.; Schroeder, S.; Kruger, N.; Herrler, T.; Erichsen, S.; Schiergens, T. S.; Herrler, G.; Wu, N. H.; Nitsche, A.; Muller, M. A.; Drosten, C.; Pohlmann, S. SARS-CoV-2 Cell Entry Depends on ACE2 and TMPRSS2 and Is Blocked by a Clinically Proven Protease Inhibitor. Cell 2020, 181, 271-280 e278, https://doi.org/10.1016/j.cell.2020.02.052.
(4) Zhou, P.; Yang, X. L.; Wang, X. G.; Hu, B.; Zhang, L.; Zhang, W.; Si, H. R.; Zhu, Y.; Li, B.; Huang, C. L.; Chen, H. D.; Chen, J.; Luo, Y.; Guo, H.; Jiang, R. D.; Liu, M. Q.; Chen, Y.; Shen, X. R.; Wang, X.; Zheng, X. S.; Zhao, K.; Chen, Q. J.; Deng, F.; Liu, L. L.; Yan, B.; Zhan, F. X.; Wang, Y. Y.; Xiao, G. F.; Shi, Z. L. A pneumonia outbreak associated with a new coronavirus of probable bat origin. Nature 2020, 579, 270-273, https://doi.org/10.1038/s41586-020-2012-7.
(5) Tian, Y.; Sun, J.; Hou, X.; Liu, Z.; Chen, Z.; Pan, X.; Wang, Y.; Ren, J.; Zhang, D.; Yang, B.; Si, L.; Bi, Y.; Liu, K.; Shang, G.; Tian, W. X.; Wang, Q.; Gao, G. F.; Niu, S. Cross-species recognition of two porcine coronaviruses to their cellular receptor aminopeptidase N of dogs and seven other species. PLoS Pathog 2025, 21, e1012836, https://doi.org/10.1371/journal.ppat.1012836.
(6) Liu, Y.; Hu, G.; Wang, Y.; Ren, W.; Zhao, X.; Ji, F.; Zhu, Y.; Feng, F.; Gong, M.; Ju, X.; Zhu, Y.; Cai, X.; Lan, J.; Guo, J.; Xie, M.; Dong, L.; Zhu, Z.; Na, J.; Wu, J.; Lan, X.; Xie, Y.; Wang, X.; Yuan, Z.; Zhang, R.; Ding, Q. Functional and genetic analysis of viral receptor ACE2 orthologs reveals a broad potential host range of SARS-CoV-2. Proc Natl Acad Sci U S A 2021, 118, https://doi.org/10.1073/pnas.2025373118.
Reviewer 2 Report
Comments and Suggestions for Authors
The authors have written a review paper on “Receptor Recognition Mechanisms in Coronaviruses.” Addressing the following items can improve the paper and make it more impactful:
-The review provides an excellent structural and comparative overview of coronavirus–host receptor interactions, however, it would benefit from a stronger conceptual synthesis for example, integrating how receptor evolution informs viral tropism, host adaptation, and future zoonotic spillovers beyond descriptive listings.
-The methodology section (Section 3.1) is comprehensive, but it can be refined for coherence and flow, emphasizing how different experimental and computational approaches ( like Cryo-EM, CRISPR, AlphaFold2) complement each other. A table summarizing advantages, limitations, and examples of each technique would add significant clarity.
-While the paper thoroughly covers α-, β-, γ-, and δ-coronaviruses, the discussion could be enhanced by highlighting translational perspectives for instance, how receptor insights guide therapeutic targeting, vaccine design, or predictive modeling of emergent coronaviruses.
Minor language and stylistic revisions are recommended for scientific precision and readability such as consistent use of tense, abbreviation formatting (like CoV, RBD, CTD), and reduction of redundancy between introductory and discussion sections. Figures 1–4 are informative but could use improved labeling and concise legends for easier cross-referencing within the text.
Comments on the Quality of English LanguageEnglish language of the paper needs to be improved.
Author Response
Comments 1. The methodology section (Section 3.1) is comprehensive, but it can be refined for coherence and flow, emphasizing how different experimental and computational approaches ( like Cryo-EM, CRISPR, AlphaFold2) complement each other. A table summarizing advantages, limitations, and examples of each technique would add significant clarity.
Response 1: Thank you for your suggestion. We now add a table summarizing advantages, limitations, and examples of each technique (like Cryo-EM, CRISPR, AlphaFold2) in the revised manuscript. Please see page 17, Table 1.
Comments 2. While the paper thoroughly covers α-, β-, γ-, and δ-coronaviruses, the discussion could be enhanced by highlighting translational perspectives for instance, how receptor insights guide therapeutic targeting, vaccine design, or predictive modeling of emergent coronaviruses.
Response 2: Thank you for the comment. We now added the translational perspective regarding how receptor insights guide therapeutic targeting, vaccine design, or predictive modeling of emergent coronaviruses. Please see the discussion part in page 15, line 690-700.
Comments 3. Minor language and stylistic revisions are recommended for scientific precision and readability such as consistent use of tense, abbreviation formatting (like CoV, RBD, CTD), and reduction of redundancy between introductory and discussion sections. Figures 1–4 are informative but could use improved labeling and concise legends for easier cross-referencing within the text.
Response 3: Thank you for the suggestion. We have checked all-through in the manuscript. We have improved the English langue in the revised manuscript. The mistakes of tense use and abbreviation formatting have been corrected. For the introduction and discussion sections, we have improved the redundancy. We have removed the improved labeling of Figures 1-4, and concise legends within the text in the revised manuscript.
Comments 4. Comments on the Quality of English Language
English language of the paper needs to be improved.
Response 4: Thank you for the suggestion. We have improved the English langue in the revised manuscript.
Round 2
Reviewer 1 Report
Comments and Suggestions for Authors
All issues have been addressed.